# A Brief Review of Nutraceutical Ingredients in Gastrointestinal Disorders: Evidence and Suggestions

**DOI:** 10.3390/ijms21051822

**Published:** 2020-03-06

**Authors:** Xiang Gao, Jingwen Liu, Li Li, Wei Liu, Meiyan Sun

**Affiliations:** 1College of Laboratory Medicine, Jilin Medical University, Jilin 132013, China; xgao15@uh.edu (X.G.); jliu79@Central.UH.EDU (J.L.); liuweijldx@sina.com (W.L.); 2Department of Pharmacological and Pharmaceutical Science, College of Pharmacy, University of Houston, Houston, TX 77004, USA

**Keywords:** functional gastrointestinal disorders, structural gastrointestinal disorders, nutraceuticals, clinical trial

## Abstract

The dietary effect on gut health has long been recognized through the empirical practice of soothing gastric discomfort with certain types of food, and recently the correlation between specific diets with lower incidences of several gastrointestinal diseases has been revealed. Ingredients from those considered beneficial foods have been isolated and studied, and some of them have already been put into the supplement market. In this review, we focus on latest studies of these food-derived ingredients for their proposed preventive and therapeutic roles in gastrointestinal disorders, with the attempt of drawing evidence-based suggestions on consuming these products.

## 1. Introduction

The relevance of diet in disease prevention has long been noticed. From the jingle saying ‘An apple a day keeps doctor away’, to modern science supported studies, people’s opinions have been shaped about what a healthy eating should be look like. For instance, limited sugar and salt intake is a key to help with general good health [1,2]. As for the well-being of the gut, fruit, vegetable and yogurt intake have been suggested for gut health promotion [3,4,5]. Now, dietary supplements have become a fast-growing market, which includes various kinds of products from vitamins to single ingredients from botanic extracts, to blends of extracts and ingredient from plants and food. Nutraceutical, as a term mixed from ‘nutrition’ and ‘pharmaceutical’, has been defined as “a food (or part of a food) that provides medical or health benefits, including the prevention and/or treatment of a disease” [6]. In terms of nutraceuticals being categorized as dietary supplements in the US, there has been a long-lasting debate about the inadequacy of their regulation. Unlike a drug, a nutraceutical product can skip the trial phase before it hits the market, from which time point, under the regulation of Dietary Supplement Health and Education Act (DSHEA) of 1994, the FDA may intervene on any concern about the product’s safety and promotional claims [7]. Hence controversy persists focusing on product efficacy and safety, as they may be compromised under several circumstances, such as limited understanding of interaction between ingredients from a whole botanic extract, or lack of sufficient tests and trials, or even the tainting of drug ingredients.

Gastrointestinal (GI) disorders can be functional or structural, with functional GI disorders as the more common type among the two [8], and both types of disorders may cause severe impairment of life quality and psychological wellbeing, and even shortened life expectancy [9,10,11]. Many nutraceutical products have appeared on the market proposed as beneficial for gastrointestinal disorders. In this review, we make our attempt to evaluate several commonly seen products based on a standard established from the strength of available evidence (Figure 1).

## 2. Role of Nutraceutical Ingredients in Gastrointestinal Disorders

### 2.1. Functional Gastrointestinal Disorders

Functional gastrointestinal disorders refer to conditions of normal body activities such as the sensitivity of the nerves in the intestines, the movement of the intestines and the way that the brain controls the intestines, which are impaired while no structural abnormalities can be observed by endoscopy, x-ray, or other tests [12]. They are the most common problems affecting the GI tract. About one quarter of people in the U.S. have one of the functional gastrointestinal disorders [13].

Functional GI disorders are a group of disorders of gut-brain interaction related to altered mucosal and immune function, altered gut microbiota, motility disturbance, and visceral hypersensitivity. Constipation, irritable bowel syndrome (IBS) and functional dyspepsia are the common examples. Nutraceutical ingredients for different GI disorders are summarized in Table 1.

#### 2.1.1. Constipation

Constipation is a common gastrointestinal disorder that refers to inadequate bowl movement or hardness of passing intestinal contents [42]. Difficulty in defecation is one of the commonly seen symptoms associated with hard and dry stool [43]. Persistence of constipation can last for weeks or even longer, which may raise the necessity of medical intervention.

Substances that either loosen stools or stimulate a bowel movement are called laxatives. For occasional constipation, laxatives are usually recommended for helping defecation. Lots of botanic products can promote this beneficial effect, and have been used even from ancient times.

Senna is one of the mostly used therapies which comes from an important class of botanic laxatives, anthraquinone drugs. The class also has cascara, frangula, aloe, and rhubarb included, and is enriched in corresponding plants and herbs in forms of glucoside derivatives of anthracene, i.e., the anthraquinones [15]. The anti-constipation effect of anthraquinone drugs comes from alteration of motility patterns and increasing of colonic fluid volume [14]. As some anthraquinone derivatives approved by FDA, the efficacy of senna or other anthraquinone containing botanic extracts can be expected. It was doubted whether the consumption of senna might associate with an increasing risk of colon cancer [44], though studies on rodents are in disagreement [45,46]. Cascara, like senna, is a type of anthranoid containing laxative. Cascara contains anthraquinone glycosides (such as cascarosides A, B, C, and D) and a small amount of anthraquinone glycosides. Several clinical trials have reported that cascara can improve stool frequency and consistency [15,16,17]. Psyllium is a frequent bulk laxative for constipation. Psyllium was demonstrated to increase stool frequency and improve stool consistency [47]. However, psyllium has its disadvantages. For instance, patients may lose appetite and have a of delay gastric emptying if taking psyllium before meals. In addition, its texture influences the compliance [48]. A randomized clinical trial with 72 subjects has shown that mixed fiber from fiber supplements is equally efficacious in improving constipation compared with psyllium. Mixed fiber is well tolerated and more effective in alleviating flatulence and bloating [18].

#### 2.1.2. Irritable Bowel Syndrome (IBS)

IBS is a functional bowl disorder that features with abdominal pain or discomfort and defecation disrupt [49]. Comparing with other G.I. disorders, the special part of IBS is that it is commonly associates with psychological conditions like depression, anxiety, and somatization. It is reported that around 60% of IBS patients have major psychological problems while etiology of this disease remains unclear [50,51,52]. Despite the etiology of IBS remains to be elucidated, manipulation of gut microbiota has been considered capable of reducing symptoms and improving patient’s life quality [53].

*Saccharomyces cerevisiae* (*S. cerevisiae*) is widely used in baking and brewing as a yeast species. Some *S. cerevisiae* strains are also taken as probiotics aiming to benefit IBS symptoms. Under current understanding, mechanisms of the beneficial effects provided by probiotics may include production of chelate degradation enzymes and short chain fatty acids, pathogenic toxins degradation and support of intestinal immunity [54]. *S. cerevisiae* strain I-3865 showed significant improvement of abdominal pain and bloating symptoms in the constipation subgroup of IBS patients [19]. Efforts have been made to discover and develop more potent *S. cerevisiae* strains for probiotic use [55], however, it is worthy to note the use of probiotic yeast strains caused rare cases of invasive *S. cerevisiae* infection, mostly in patients with immunosuppressive conditions [56,57]. The symptom improving expectation on *S. cerevisiae* consumption may depend on specific strains, and patients with compromised immune function should use the product with caution.

*Andrographis paniculata* is a plant growing mainly in south and southeastern Asia. It has demonstrated a significant decrease of cytokines mRNA expression, including TNF-α and IL-6 [58], both of which have increased levels in patients with IBS [59]. Mice with severe intestinal inflammation receiving *Andrographis paniculata* showed inhibited development of chronic colitis [20]. Additionally, andrographolide-loaded nanoemulsion (AG-NE) formulation demonstrated an enhanced oral bioavailability and anti-inflammatory activity [60].

Boswellic acids are active components of *Boswellia serrata*. More than 12 different boswellic acids have been identified so far [61]. They have been demonstrated anti-inflammatory activity by inhibition of leukotriene synthesis and prostaglandin synthesis [62]. When a lecithin-based delivery form of boswellic acids was given to healthy participants with a mild form of IBS compared with standard management group, it was found that the *Boswellia* group showed a significant lower need for rescue medications and consultations or medical evaluation [21]. Although the treatment was effective and demonstrated safety, subjects in treatment group were more likely to experience adverse events than those treated with standard management. Belcaro et al. has reported the same results [63].

*Curcuma longa* is a type of herb belonging to ginger family, which is known for the antioxidant, anti-inflammatory, anticancer properties and the ability to modulate gut microbiota [22]. Curcumin has also been demonstrated to be safe and well-tolerated in the available trials [64]. Random-effects meta-analysis based on three studies and 326 patients found that curcumin have a beneficial effect on IBS symptoms [22].

Another study evaluated a mixture of several herbal medicines (*Boswellia carterii, Zingiber officinale*, and *Achillea millefolium*) with positive effects in patients with IBS [23]. The results showed that the mean score of abdominal pain severity and frequency, bloating score, and depression and anxiety score were statistically significant decreased in patients administered herbal medication compared with control group.

Butyrate analogue/derivatives have been used in trials to study their beneficial potent in various diseases like cancer or hematopoietic deficiencies [65,66]. Focusing on gut health, butyrate, as the most emphasized ingredient of short chain fatty acids (SCFAs), was studied in different trials that came out with contradicting suggestions [67]. Naturally, a major source of the SCFAs is the secondary product of food that goes through microbiota fermentation in the human colon, and the intrinsic complexity of the nutrient–microbiota–colon network could be a reason for this divergence. Direct dosing of butyrate showed improvment in symptoms including abdominal pain, meteorism and flatulence in patients with the IBS-diarrhea subtype (IBS-D) [24]. Latest studies in animals revealed butyrate may exert these beneficial effects through the modulation of AMP-activated protein kinase (AMPK) [25].

#### 2.1.3. Functional Dyspepsia

Functional dyspepsia (FD) is one of the most prevalent chronic functional gastrointestinal disorder which includes three subtypes: postprandial distress syndrome (PDS), epigastric pain syndrome (EPS) and a subtype with both PDS and EPS features [68]. The pathogenesis remains unclear and is likely multifactorial. The underlying causes include excessive acid secretion, inflammation of the stomach, food allergies, psychological factors and side effects caused by medications. The common symptoms are unexplained fullness and bloating after eating, early satiety, excessive burping, and epigastric pain or burning. Currently, neutralizing acid and blocking its production are the two main methods to treat functional dyspepsia. However, those treatment drugs have severe side effects including cardiac arrest and sudden death [69,70]. Botanicals could be a promising source of natural prokinetics which can address this issue.

Nannoni et al. demonstrated a two-step process to extract a relatively high concentration of actives from *Perilla frutescens* leaf. The *Perilla frutescens* leaf extract is a new botanical endowed with prokinetic, anti-spasmodic and anti-inflammatory properties which is a promising candidate for the treatment of FD [28].

*Pimpinella anisum* (Apiaceae), also known as aniseed, is one of the oldest species used in Egypt, Greece, Rome, and the Middle East. Its essential oil has been reported to have antispasmodic, secretolytic, secretomotor, and antibacterial effects [71]. Triapelli et al. have justified that extracts of *Pimpinella anisum* activated the NO-cGMP pathway and had a significant muscle relaxant effect [72]. A double-blind, randomized clinical trial have confirmed the effects of *Pimpinella anisum* on relieving the symptoms of functional dyspepsia [29].

#### 2.1.4. Acid Reflux

Acid reflux, also known as gastroesophageal reflux disease (GERD), is the back flow of stomach acid into esophagus that causes the primary symptom as heartburn. It is one of most common digestive problems seen in daily life, and can be occasionally provoked by irritating foods like coffee or cold drinks [73]. Persistence of the condition is called gastroesophageal reflux disease (GERD) and may lead to symptoms including acid taste in the mouth, heartburn, bad breath, chest pain, regurgitation, and breathing and teeth problems, etc. [74].

Alginic acid is a polysaccharide from brown seaweed. It has been used for many years in food, cosmetics, and pharmaceutical products as an approved ingredient by FDA, as emulsifier, thickener, and stabilizer [75]. As a common formulating material in antacid drugs, itself has been claimed to also contribute soothing the reflux symptom by forming a gel ‘raft’ floating on top of the gastric contents, which can entrap carbon dioxide (CO_2_) and antacid components contained to form a protective pH neutral barrier [76]. A trial in pregnant women shows that, compared with plain magnesium-aluminum antacid gel, alginic acid formulation failed to make a difference in symptom controlling and life quality improving [30]. This suggests the solo use of alginic acid for alleviating reflux symptoms may have efficacy concerns in certain physiological conditions. Based on the above, it is suggested alginic acid should always to be taken with antacids but not alone when used against reflux for symptom alleviation.

*Aloe vera* (A. vera) gel is an extract of *Aloe barbadensis,* which has wide applications for the treatment of GI disorders. A. vera gel has been reported to possess biological effects such as wound-healing [77], antimicrobial effects [78], anti-inflammatory [79], and anti-proliferation [80]. All of these properties are important for the treatment of GERD. In a randomized, positive controlled trial, 4 weeks of A. vera syrup administration to subjects showed A. vera was safe, well tolerated and effective for reducing the symptoms of GERD [31].

### 2.2. Ingredients for Structual Gastrointestinal Disorders

Ingestion and digestion of dietary substances are two of the key function of the gastrointestinal system, which is responsible for the absorption of nutrients, and the elimination of waste products [81]. The most important structures of the gastrointestinal system are the intestinal epithelial barrier (IEB) that can prevent the toxic contents while transport contents between blood and intestinal lumen [82]. Studies have shown that gastrointestinal dysfunction can cause many disorders including metabolic and inflammatory diseases [83,84]. Normally, those diseased are caused by functional or structural gastrointestinal disorders. Here, the discussion will focus on the structural disorders.

A structural gastrointestinal disorder is refered to gastrointestinal abnormalities when the internal structure or organ looks abnormal and does not function properly. Usually, the gastrointestinal disorders come with one or more of those symptoms including abdominal pain, diarrhea, bleeding and etc. [85,86]. It can directly or indirectly caused by physiological factors like anxiety and depression, and biological factors, comprising abnormalities in GI motility, mucosal and immune function [87]. The structural gastrointestinal disorder is typically easy to be diagnosed by a more in-depth diagnosis like endoscopic surveillance [88]. Sometimes, the structural abnormality needs to be removed surgically. The most common examples of structural gastrointesinal disorders include inflammatory bowel disease, diverticular disease, hemorrhoids, colon polyps and colon cancer.

#### 2.2.1. Inflammatory Bowel Disease (IBD)

IBD is a class of chronic inflammatory gastrointestinal diseases, with ulcerative colitis (UC) and Crohn’s disease (CD) as the two major types [89]. Various factors may contribute to the incidence of IBD, including environmental factors, infectious diseases, ethnicity, genetic susceptibility and dietary habits. The progression of IBD can also increase the risk of colorectal cancer [90,91]. Nutraceuticals, mainly probiotics and phytochemicals, are proposed to useful in controlling the symptoms and lower risks of malignancy development.

As belonging to flavonoids, anthocyanins can be found in various kinds of plants and herbs. They are actually water-soluble pigments and abundant in diet like blueberry, raspberry, bilberry, black rice and black beans, etc. Anthocyanins were shown capable of inhibiting pro-inflammation molecules and pathways [32], which were considered contributing the anti-inflammatory effect and alleviation of tissue damage in UC patients [33]. As substantial trials are yet to be cast out in IBD or colon cancer patients, it is still inconclusive for deciding whether anthocyanins should be taken to benefit the disease.

Curcumin is another flavonoid compound that derives from turmeric, and is a commonly seen product in the supplements market. Curcumin has multiple pharmacologic activities which may deliver benefits by its antioxidant, anti-inflammatory, antitumor, immunomodulatory, wound-healing, anti-proliferative, and antimicrobial activities [34]. Combinational use of curcumin with other drugs in UC patients gained better therapeutic effect, and clinical activity index in curcumin treating patient groups were significantly improved in multiple trials [92,93,94,95]. Base on the above, consumption of curcumin may be good for IBD patients.

#### 2.2.2. Gastrointestinal Neoplasia

Gastrointestinal neoplasia is a general name of the malignant disorders occur in gastrointestinal tract and its accessory organs. Among all types of cancer, gastrointestinal neoplasia is common in population all over the world and is highly associated with life-styles, which is believed can be actively prevented through appropriate lifestyle modification including dietary habits [96].

Gamma-aminobutyric acid (GABA) is a natural non-coding amino acid that broadly exists in many plant parts, as well as in the human body. Extraneous GABA showed anti-proliferative effect in colon cancer cells by interfering key cancer cell signaling pathways, and suppressed tumor growth in animal models [35]. However, apart from studies concerning GABA’s neuromodulator roles in relevant conditions, trials concerning potential therapeutic roles of GABA in gastrointestinal neoplasia are yet to be cast out.

Resveratrol is a polyphenol compound, similar to flavonoids like curcumin, which has been shown to target pro-inflammatory molecules and pathways in in vivo studies [97,98,99]. Studies also suggested the anti-inflammatory effect of resveratrol may further contribute to tumor-prevention and suppression effects in animal models [98,99,100]. Clinical trials in colorectal cancer patients gave out certain indications that the compound was, at least, presented pharmacological effects as elevation of apoptotic markers in cancer cells [36,37]. However, these trials were yet to reach end points that could bring out the conclusion of resveratrol’s clinical utility.

#### 2.2.3. Diverticular Disease

Diverticular disease of the colon is one of the most common disorders with increasing prevalence in western countries [101]. Diverticular disease is a medication status and characterized by the outpouching of the colonic mucosa and submucosa through the muscular layer [102]. Diverticular disease has asymptomatic diverticulosis and symptomatic diverticulitis. Usually about 20% patients with diverticular disease experience symptomatic diverticulitis like recurrent abdominal pain or rapture [103,104]. The disease risk factors include age, diet, gut microbiota, genetic predisposition, environmental factors and colonic dysmotility. Nowadays, the most effective treatments for diverticular disease are implicated to fiber, non-absorbable antibiotics and probiotics [105]. Unfortunately, the medical strategy to prevent diverticulitis recurrence is still limited. Herein, we discuss some ingredients which are beneficial to diverticular disease and might help to prevent it.

Psyllium is a commonly used soluble dietary fiber from the husks of the psyllium (*Plantago ovata*) seed, associated with a potential role in the treatment and prevention of bowel diseases such as diverticulosis, irritable bowel syndrome and inflammatory bowel disease [106]. Soluble fiber like psyllium and ispaghula will be fermented in the colon by bacteria when dissolving in water and forming a gel. The metabolites, short chain fatty acids and gas were reported to shorten the gut transit time and alleviate constipation and intracolonic pressure [107]. Jalanka’s studies suggested that psyllium supplementation altered the physiological environment and was associated with the microbiome composition which may improve bowel function and gastrointestinal symptoms [38]. Marilia also illustrated that the soluble fibers like psyllium were beneficial for patients with colonic diverticula. However, he mentioned that more effort is needed to prove the benefit of dietary of fiber for diverticular patients due to the substantial methodological limitations and the lack of ad hoc designs in current studies [108].

Quercetin is a well-known polyphenol which is widely distributed in many fruits and plants like apples, onion, green tea, etc. [109]. Quercetin undergoes extensive phase II metabolism in the intestine and liver and presents as different forms of its metabolites [110]. Moreover, quercetin is claimed to exert many biological functions against allergies, inflammation, microbes, ulcers, hepatotoxin, viruses and tumors [111]. Torras et al. demonstrated that plant extraction may have some nonspecific involvement with proteins in bacteria cell walls, which inactivated enzymes and affect proteins transport [112]. Kamil Sierzant’s studies indicated that the feeding of broilers with a mixture of quercetin and other plants led to a reduction of microorganisms in the intestine, which showed that the quercetin extraction contributes to the prevention of diverticular disease [39]. Although most in vitro studies suggests that quercetin has anti-inflammation and immunological effect, the results from a double-blinded, placebo-controlled, randomized trial demonstrated that quercetin supplementation did not have effect on the innate immune system in community-dwelling adult females at 500 and 1000 mg/day for 12 weeks.

### 2.3. Colon Cancer

Colorectal cancer (CRC), also known as bowel cancer, colon cancer, or rectal cancer, is the third most common malignancy in the United States [113]. Colorectal cancer usually begins with polyps, a nonspecific term to describe a unknown growth on epithelial wall of the colon [114]. Normally, polyps are often non-cancerous growths, but some can develop into cancer if remain untreated, thus invading the colonic wall. Presently, the colonoscopy is the most common tool to diagnose the disease and evaluate the effectiveness of therapies, but this is invasive, time consuming, and expensive, which precludes continuous monitoring [115]. Scientists have put many efforts into colon cancer preventions and treatments, and there are some ingredients reported to be beneficial for preventing colon cancer developments.

Omega-3 and omega-6 fatty acids are the two major classes of polyunsaturated fatty acids (PUFAs), characterized by the presence of a double bond three atoms away from the terminal methyl group in their chemical structure. Those essential fatty acid could not be formed by our body and have to be gained from certain foods such as flaxseed and fish, as well as dietary supplements such as fish oil [116]. Recent studies indicated that omega-3 showed anti-inflammatory effect by antagonizing arachidonic acid-induced pro-inflammatory prostaglandin (PGE2) formation, which was a known compound that can cause colon cancer [117,118]. Another well-known mechanism underlying the anti-inflammation effects of omega-3 was blocking the activation of the pro-inflammatory transcription factor nuclear factor kB thus alleviating the inflammation and preventing the colon cancer development [40]. However, although omega-3 is beneficial to human health, the secondary lipid oxidation products, especially in the presence of heme iron or pro-oxidants, are potential hazardous and so it is suggested to be used with caution [119].

Folic acid and folate are two forms of vitamin B, which are essential for DNA synthesis, repair, methylation and aberrations of which contribute to the development of colorectal cancer [120]. Vitamin B is naturally present in in dark green leafy vegetables, beans, peas and nuts [121]. Folate and folic acid are rich in fruits including oranges, lemons, bananas, melons and strawberries. The randomized controlled trial conducted at Dublin hospital from O’Reilly’s study demonstrated that decreased vitamin B level was associated with increased risks of colon cancer [122]. It was reported that the inverse association between vitamin B level and risk of colon cancer was related to genomic instability (mis-incorporated uracil, DNA single strand breakage and DNA repair capacity) in response to vitamin level (deficient or supplemented) in different in-vitro cell studies, rodent models and human case-control studies [123]. However, the intervention time of vitamin B supplementation on colorectal cancer risk was suggested to be an essential factor that can affect the modulatory effect [41].

## 3. Safety and Efficacy of Nutraceuticals

For drug development process, it is essential to have preclinical research on animals and clinical studies on humans in order to illustrate the therapeutic effects. In the case of nutrition, there was no verification method to prove preventing diseases in the past. In recent years, nutraceuticals have been proven by researchers to influence diseases such as cancers, heart diseases, high cholesterol and hypertension. As many nutraceuticals have been applied as alternatives for medicine, the regulation of these products is cause for social attention.

For nutraceutical products, evaluation of safety is usually easier to perform than efficacy. Manufactures should have solid data and concrete proof to support the claim of safety and efficacy of their products. Consumers need assurance that a product is safe and has the effects that correspond to how they are labelled. Currently, the Food and Drug Administration (FDA) regulates them in the same way with all foods. The scientific and regulatory challenges exist in research on the efficacy of nutraceutical products, which are common to all countries worldwide. As the market has expanded, so has the amount of scientific studies and research to support or oppose their use. It becomes more complex when there is conflicting evidence and conflicting ways for evaluating them. The European Food Safety Authority (EFSA) had a negative opinion about most of the requests of authorization to the use of health claims related to the GI tract, mainly due to an insufficient substantiation of the claimed effect [124]. More questions are needed to be addressed. For instance, how the traditional evidence be evaluated within the framework of traditional healing theories; what should be done if randomized clinical trials do not support the traditional healing; and how the consumers to make choices if they want to explore both conventional and traditional medicine. Above anything else, the safety of nutraceuticals need to be guaranteed for humans.

## 4. Conclusions and Future Directions

GI disorders remain prevalent and difficult-to-manage disease conditions. The nutraceuticals have demonstrated biological effects by involving the biochemical reactions as substrates, cofactors and inhibitors of specific enzymes or receptors. There is growing interest in the use of these nutraceuticals to manage gastrointestinal disorders. Generally, nutraceuticals are considered to be safer than conventional pharmaceutical therapies, which encourages the patients to apply alternative options to alleviate symptoms of GI disorders.

In this review, we looked at several nutraceutical ingredients, and preferably on the well-characterized, single component products. Reasons for being skeptical on ingredients, especially blends or whole plant extracts, are related to identification of the active pharmaceutical ingredient and the yet to be understood onset mechanisms. For example, soy protein can lower blood cholesterol in the human body [125], though the debate was on whether it was the protein itself, or the fiber contained, or some other ingredients that were responsible for the effect. Different suggestions were made, like saponins [126], or isoflavones [127], both supported by studies, which indicates a fact that active ingredients from food may, and could be very likely, interact to deliver those good effects as we are expecting. A thorough chemical and biological characterization of active ingredients that may interact is hard, and usually yet to be finished for a clear interpretation of using a certain ingredient, even if it is already on the market. Although growing evidence showing nutraceuticals may help to treat diseases like GI disorders, there is still a long way to go before putting nutraceuticals as considerable alterations or complementation of drugs, and efforts should be made pivotally in the establishment of industrial standardization and adequate, more active product regulation.

## Figures and Tables

**Figure 1 ijms-21-01822-f001:**
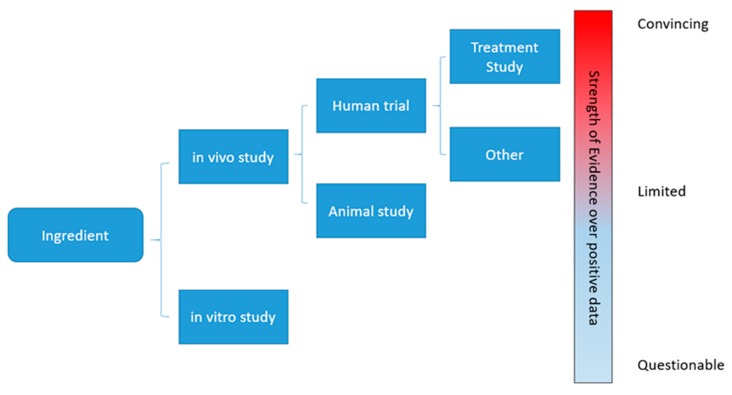
Strength of evidence levels for evaluation.

**Table 1 ijms-21-01822-t001:** Nutraceuticals for Gastrointestinal disorders.

Reference	Nutraceuticals	Categories	Gastrointestinal Disorders	Study Object	Outcomes
[14]	Senna	TCM/Anthraquinone	Constipation	human	Anti-constipation effect
[15,16,17]	Cascara	TCM/Anthraquinone	Constipation	human	Improve stool frequency and consistency
[18]	Mixed fiber	Fiber	Constipation	human	Alleviate flatulence, bloating
[19]	*Saccharomyces cerevisiae*	Yeast	IBS	human	Improve abdominal pain and bloating symptoms
[20]	*Andrographis paniculata*	TCM	IBS	human	Inhibit development of chronic colitis
[21]	Boswellic acids	Pentacyclic triterpene	IBS	human	Prevent symptoms of IBS
[22]	*Curcuma longa*	TCM	IBS	human	Beneficial effect on IBS symptoms
[23]	Herbal mixture of *Boswellia carterii, Zingiber officinale,* and *Achillea Millefolium*	TCM	IBS	human	Positive effects
[24,25]	Butyrate	short chain fatty acids (SCFAs)	IBS	Animal (rat)	Alleviate inflammation and pain in IBS models
[26,27]	Purple potato extracts	Polyphenols	IBS	Animal (mouse)	Improved colitis symptoms, colonic structure distortion, and inflammation
[28]	*Perilla frutescens* leaf extract	TCM	FD	human	Anti-spasmodic and anti-inflammatory
[29]	Apiaceae	TCM	FD	human	Relieve the symptoms of FD
[30]	Alginic acid	Polysaccharide	Acid reflux	human	Alleviate reflux symptoms
[31]	*Aloe vera* gel	TCM	Acid reflux	human	Effective for reducing the symptoms of GERD
[32,33]	Anthocyanins	Flavonoid	IBD	Ex-vivo (human)	Anti-inflammatory effect and alleviation of tissue damage in UC patients
[34]	Curcumin	Flavonoid	IBD	human	Improved clinical activity index
[35]	GABA	Amino acid	Colon cancer	Animal (mouse)	Anti-proliferative effect
[36,37]	Resveratrol	Phenol	Gastrointestinal neoplasia	human	Target pro-inflammatory molecules and pathways
[38]	Psyllium	Fibre	Diverticular disease	human	Improve bowel function and gastrointestinal symptoms
[39]	Quercetin	Flavonoid	Diverticular disease	Animal (chicken)	Prevent diverticular disease
[40]	Omega-3 (n-3) polyunsaturated fatty acids	Polyunsaturated fatty acids	CRC	human	Alleviate the inflammation and preventing the colon cancer development
[41]	Vitamin B	Water-soluble vitamins	CRC	human	Reduce the risk of CRC

IBS: irritable bowel syndrome; FD: functional dyspepsia; GERD: gastroesophageal reflux disease; IBD: inflammatory bowel disease; GABA: gamma-aminobutyric acid; CRC: colorectal cancer; UC: ulcerative colitis; TCM: traditional Chinese medicine.

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
