# Peer review of "A Brief Review of Nutraceutical Ingredients in Gastrointestinal Disorders: Evidence and Suggestions"

_ijms, 2020, doi:10.3390/ijms21051822_

Round 1

Reviewer 1 Report

This review aims to summarize the dietary intervention on gastrointestinal health and diseases. The authors briefly discussed some beneficial effects of nutraceuticals on functional/structural disorders. Although this study is an interesting research topic, the comprehensive and in-depth discussion need to be included in this manuscript. The authors could consider narrow down the topic or add more discussion with detailed information.

  1. All conclusions are from human, animal, or cell studies?
  2. What are the categories of listed nutraceuticals? For instance, are they belong to polyphenols or short-chain fatty acids?
  3. More literature publications are needed. For example, purple potato and short-chain fatty acids have been reported to regulate the intestinal health via a metabolic enzyme, AMP-activated kinase. This enzyme then mediates the intestinal barrier function via transcriptional factors.

Author Response

Dear Reviewer,

Thank you.

We appreciate all the comments and suggestions from you, and we have revised the manuscript accordingly. The modifications can be tracked in the updated manuscript. Please check the response/list of changes here below.

Best regards,

Xiang Gao

=================================================

Reviewer 1:

  1. All conclusions are from human, animal, or cell studies?

Response: Thank you for the suggestion. We have added a new column (categories) to provide address this question. The study object of each reference has been listed. Please see table 1 on page 6-7.

  1. What are the categories of listed nutraceuticals? For instance, are they belong to polyphenols or short-chain fatty acids?

Response: Thank you for the suggestion. We have added a new column (categories) to provide more information about the listed nutraceuticals. Please see table 1 on page 6-7.

  1. More literature publications are needed. For example, purple potato and short-chain fatty acids have been reported to regulate the intestinal health via a metabolic enzyme, AMP-activated kinase. This enzyme then mediates the intestinal barrier function via transcriptional factors.

Response: We have added more literature in the reversed manuscript. The following paragraph has been added to pages 4-5. Two new rows have been added to table 1 on page 7. And seven more citations (reference 121~127) have been added.

Butyrate analogue/derivatives have been using in trials to study their beneficial potent in various diseases like cancer or hematopoietic deficiencies (125, 126). Focusing on gut health, butyrate, as the most emphasized ingredient of short-chain fatty acids (SCFAs), was studied in different trials that came out with contradicting suggestions (127). Naturally, a major source of the SCFAs is the secondary product of food that goes through microbiota fermentation in the human colon, and the intrinsic complexity of nutrient - microbiota – colon network could be a reason to this divergence. Direct dosing of butyrate showed improvement in symptoms including abdominal pain, meteorism, and flatulence in patients with the IBS-Diarrhea subtype (IBS-D) (121). The latest study in animal revealed butyrate may exert these beneficial effects through the modulation to AMP-activated protein kinase (AMPK) (122).

Meanwhile, we replaced reference #17, and changed the description of the nutraceuticals for ref.46 in the table.

Reviewer 2 Report

The manuscript is a narrative review aiming at summarize the effects of some “nutraceuticals” on some markers of gastro-intestinal disorders.

I think the manuscript is scientifically unsound, resulting mostly a collection of information made by traditional medicine and with some reference on studies, with no particular scientific evidences. In fact, the most important chapter, chapter 3 is barely considered. For example, Biasini et al.,  International Journal of Food Sciences and Nutrition (2018).69:771-804, reviewed the literature in order to collect, collate and critically evaluate the gold standard method for the use of specific  “nutrition and health claims” of products. In these manuscript it is evident as most of the gut discomfort health claims are not valid and the methods are not appropriate for supporting the claim. So, most of the information treated in this review comes from the popular medicine and are not supported by science. Really don’t know for eastern medicine and how the popular medicine is considered, but in the western medicine EFSA or FDA should apply the yes for the claim role, and no one of the example done has been scientifically validated. So, this is a review for folklore medicine.

For example, quercetin, which is not a polysaccharide,  has been studied for biological activity on GI pathologies, but no trial evidenced any effects.

Author Response

Dear Reviewer,

Thank you.

We appreciate all the comments and suggestions from you, and we have revised the manuscript accordingly. The modifications can be tracked in the updated manuscript. Please check the response/list of changes here below.

Best regards,

Xiang Gao

===================================================

Reviewer 2:

Comment: I think the manuscript is scientifically unsound, resulting mostly a collection of information made by traditional medicine and with some reference on studies, with no particular scientific evidences.

Response: Thank you for your comments. The information in this manuscripts were cited with proper references which are published on various journals such as Lancet (reference #1), Nature (reference #2), Cancer Research (reference #5, #88), Gastroenterology (reference #8, 12, 27), American Journal of Clinical Nutrition (reference # 118), etc. These studies are evaluated either in vitro or in vivo with informative data (we have added this information for each of the reference in the main text). For example, in reference #38, this paper demonstrated the HMPL-004 (extracts of the plant Andrographis paniculata) inhibited the development of chronic colitis by affecting early T-cell proliferation, differentiation in a murine model of chronic colitis. Please kindly pinpoint the references we cited which are not appropriate or lack of scientific proof, we would like to change it in the text. 

Comment: In fact, the most important chapter, chapter 3 is barely considered. For example, Biasini et al., International Journal of Food Sciences and Nutrition (2018).69:771-804, reviewed the literature in order to collect, collate and critically evaluate the gold standard method for the use of specific “nutrition and health claims” of products. In these manuscript it is evident as most of the gut discomfort health claims are not valid and the methods are not appropriate for supporting the claim. So, most of the information treated in this review comes from the popular medicine and are not supported by science. Really don’t know for eastern medicine and how the popular medicine is considered, but in the western medicine EFSA or FDA should apply the yes for the claim role, and no one of the example done has been scientifically validated. So, this is a review for folklore medicine.

Response: We agree that section 3 needs more discussion. Although we highlighted the importance of the safety of the nutraceuticals, the efficacy issue is not well discussed. We have rewritten this section and added this critical thinking in the manuscript (changes have been made in the revision manuscript on page 12) as the following:

The scientific and regulatory challenges exist in research on the efficacy of nutraceutical products, which are common to all countries worldwide. As the market has expanded, so has the amount of scientific studies and research to support or oppose the use. It makes more complex when there are conflicting evidence and conflicting ways for evaluating them. The European Food Safety Authority (EFSA) had a negative opinion about most of the requests of authorization to the use of health claims related to the GI tract, mainly due to an insufficient substantiation of the claimed effect. More questions are needed to be addressed. For instance, how the traditional evidence be evaluated within the framework of traditional healing theories; what should be done if randomized clinical trials do not support the traditional healing; and how the consumers to make choices if they want to explore both conventional and traditional medicine.

Comment: For example, quercetin, which is not a polysaccharide, has been studied for biological activity on GI pathologies, but no trial evidenced any effects.

Response: Thanks for the correction of quercetin’s category. We have made the modification in the text accordingly (page 10 line 262). And we did some literature research, although many in vitro studies suggested the anti-inflammation effect of quercetin supplementation, there are some articles demonstrated that it had no effect in the clinical trials. We have added this information on page 10.

Round 2

Reviewer 1 Report

Thank you for the revision. I agreed to publish this review in IJMS.

Reviewer 2 Report

Authors produced a rebuttal letter to reviewers' queries, but in my opinion the major lack remains related to the not scientific validity of the arguments treated, and I confirm this is more stuck to folklore medicine.

Having as references paper published in Journals with high impact does not mean that what authors are stating is soldi and valid.

I leave to the editor the last decision.